# Estimation of Finite Finger Joint Centers of Rotation Using 3D Hand Skeleton Motions Reconstructed from CT Scans

**Xiaopeng Yang [1], Zhichan Lim [2], Hayoung Jung [2], Younggi Hong [2], Mengfei Zhang [1], Dougho Park [3] and Heecheon You [2,*]**

[1] School of Artificial Intelligence and Computer Science, Jiangnan University, Wuxi 214122, China; yxp233@jiangnan.edu.cn (X.Y.); 6191611049@stu.jiangnan.edu.cn (M.Z.)

[2] Department of Industrial and Management Engineering, Pohang University of Science and Technology, Pohang 37673, Korea; limzchan@postech.ac.kr (Z.L.); niceterran36@postech.ac.kr (H.J.); ygh0152@postech.ac.kr (Y.H.)

[3] Department of Rehabilitation Medicine, Pohang Stroke and Spine Hospital, Pohang 37659, Korea; parkdougho@gmail.com

\* Correspondence: hcyou@postech.ac.kr

**Abstract:** The present study proposed a method to estimate the finite finger joint centers of rotation (CoRs) with high accuracy using 3D hand skeleton motions reconstructed from CT scans. Ten hand postures starting from a fully extended posture and ending at a fist posture with about 10° difference in flexion between the adjacent postures were captured by a CT scanner for 15 male participants, and their 3D hand skeletons were reconstructed using the CT scans. Each bone segment from the full extension posture was registered to the corresponding bone segments of the remaining hand postures. The proximal bone segments of a joint from two postures were aligned to estimate the finite CoR of the joint between the two postures. Centerlines of the distal bone segments of the joint were then identified using the principal component analysis method, and the finite CoR of the joint was determined as the intersection point of the identified centerlines. The proposed method reduced the variation of estimated finite joint CoRs by 16.0% to 67.0% among the finger joints compared to the existing methods. The variation of estimated finite joint CoRs decreased as the rotation angle of the joint increased. The proposed method can be used for the simulation of finger movement with high accuracy.

**Keywords:** finite center of rotation; finger joint; hand skeleton motion; CT scan

## 1. Introduction

The hand is a complex interface that performs various manual tasks, such as manipulating objects, communicating, typing, and playing musical instruments. Digital human hand models have been widely used in ergonomic product design and evaluation [1–7]. For example, Endo et al. [4] developed a system for the ergonomic design and assessment of a handheld information appliance by integrating a digital hand with a product model and corresponding tasks to save development time and cost.

Most digital hand models in the market and existing studies have been established based on the assumption that the hand is a rigid linkage system, indicating that hand segments rotate around fixed joint centers of rotation (CoRs) in the models. Fixed joint CoRs can be estimated using surface marker-defined finger motions [8–11] or bone curvature-based method [12,13]. For example, Zhang et al. [11] estimated finger joint CoR locations from measured surface marker flexion-extension motions by minimizing the time-variance of the internal link lengths based on an empirically

quantifiable relationship between the local movement of a surface marker around a joint and the joint flexion-extension angle. Fowler et al. [13] reconstructed a 3D hand skeleton from magnetic resonance imaging (MRI) scans and estimated the location of a finger joint CoR as the center of curvature of the portion of the head of the proximal bone segment in contact with the base of the distal bone segment of the finger joint in the sagittal plane.

Finite joint CoRs need to be estimated since the hand is actually a non-rigid linkage system. Reuleaux [14] proposed a geometric method to estimate the locations of finite joint CoRs by measuring the relative displacement of two adjacent hand segments. Challis [15] validated Reuleaux's method and found that the error in estimating the finite joint CoR decreased as the rotation angle of the joint increased. Based on Challis' report, to achieve relatively high accuracy (error < 2 mm) in finite joint CoR location estimation using Reuleaux's method, the rotation angle of a joint needs to be larger than 20°. Silva da et al. [16] applied Reuleaux's method to estimate the locations of finite finger joint CoRs by marking two points on the hand skin surface along the direction of the distal hand segment of a joint. Figueroa et al. [17] developed a method to estimate the locations of finite finger joint CoRs by aligning the proximal bone segments of a joint at different postures using the iterative closest point (ICP) algorithm [18] to form the relative movement of the distal bone segments of the joint. However, as shown in Figure 1, adapted from Figueroa et al.'s report, an obvious error among the estimated locations of finite finger joint CoRs appeared. The error could be caused by the differences in the 3D bone surfaces at different postures reconstructed from different computed tomography (CT) scans due to the intensity differences at different CT scans.

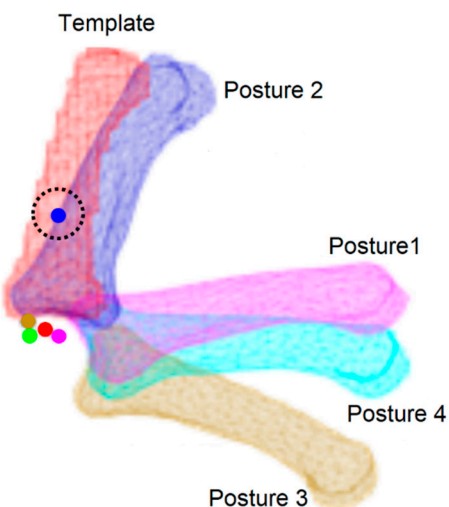

**Figure 1.** Estimated locations of finite joint centers of rotation of the proximal interphalangeal joint of the middle finger between different postures with an error highlighted in a dotted circle by Figueroa et al.'s method (adapted).

The present study proposed to use the same bone surfaces at different postures for better estimation of the locations of finite finger joint CoRs. The 3D hand skeleton at the full extension posture was selected as the hand skeleton template. After the connected bone segments in the template were separated, each bone segment in the template was registered to the corresponding bone segment at a particular hand posture. Then, the registered hand skeletons at another hand posture with the template were used for estimation of the locations of finite finger joint CoRs. Through registration, the bone surfaces at particular postures were the same as those at the template posture. The proposed method was applied to find the locations of the finite finger joint CoRs at metacarpophalangeal (MCP), proximal interphalangeal (PIP), and distal interphalangeal (DIP) joints of the index, middle, ring, and little fingers. The performance of the proposed method was evaluated through comparison with Reuleaux's and Figueroa et al.'s methods.

## 2. Materials and Methods

### 2.1. Participants

Fifteen males (age = 23.7 ± 2.0 years, ranged from 20 to 28) with various hand sizes participated in the study. Hand sizes were classified into three groups based on hand length (small: ≤ 181.0 mm for less than the 33rd percentile; medium: 181.0 to 187.0 mm for the 33rd to 66th percentiles; large: > 187.0 mm for greater than the 66th percentile; Size Korea, 2010), with five participants in each group. All the participants were right-handed and had no history of musculoskeletal injuries. The study was approved by the Institutional Review Board of Pohang Stroke and Spine Hospital, and informed consent was obtained.

### 2.2. Hand Posture Data Acquisition and Processing

CT scans of the participants' right hands in ten postures were captured by a radiologist at Pohang Stroke and Spine Hospital. The ten postures (postures 1 to 10; Figure 2) were selected from a natural hand motion starting from a fully extended posture (posture 1) and ending at a fist posture (posture 10) with a difference of approximately 10 degrees in flexion of the PIP joint of the index finger between two adjacent postures visually estimated by an experimenter. The difference of the PIP joint angles was not measured during the CT scan to avoid unnatural motions of the hand. The ten postures were sequentially scanned by a 256-slice CT scanner (Brilliance iCT; Philips Healthcare, Cleveland, OH, USA) while the participants were holding each of the postures. The participants were covered by a lead-free radiation shielding apron (FC001; Longkou Sanyi Medical Device Co., Ltd., Longkou, China) to protect them from radiation. For each posture, 576 to 670 CT slices, depending on the participants' hand sizes, were collected with a resolution of 512 × 512 pixels and a thickness of 0.44 mm for each CT slice.

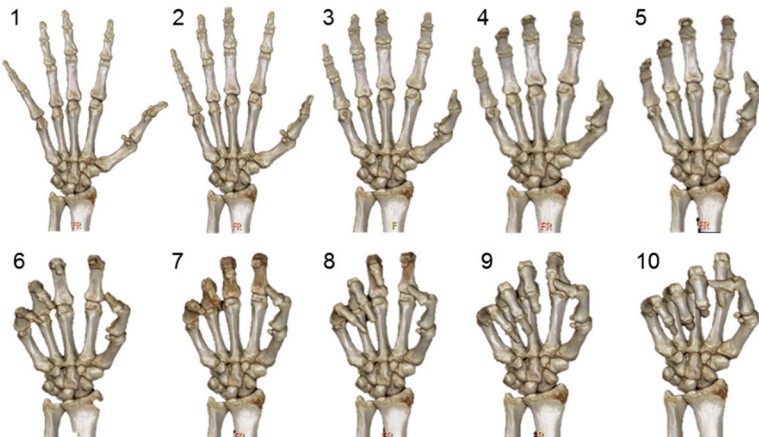

**Figure 2.** Postures selected from a natural hand motion starting from a fully extended posture (posture 1) to a fist posture (posture 10) for the estimation of finite finger joint centers of rotation.

A 3D hand skeleton was reconstructed using CT scans of each hand posture. First, the hand skin was excluded from a CT scan using a masking program developed by the Visualization Toolkit (VTK) [19], as shown in Figure 3a. Second, the hand skeleton was semi-automatically extracted from the masked CT scan by a threshold-based segmentation method provided in the Medical Imaging Interaction Toolkit (MITK) [20], as shown in Figure 3b. Lastly, the bones connected were separated from each other for the hand skeleton of posture 1 using a subtraction function in MITK, as shown in Figure 3c. After the separation of the bones, the hand skeleton of posture 1 consisted of 29 bones, including radius, ulna, eight carpal bones (hamate, pisiform, triquetral, lunate, scaphoid, capitate, trapezium, and trapezoid), five metacarpals, five proximal phalanges, four middle phalanges, and five distal phalanges. Each bone separated in the hand skeleton of posture 1 was exported as a file with the polygon (PLY) format by RapidForm 2006 (Inus Technology, Inc., Republic of Korea).

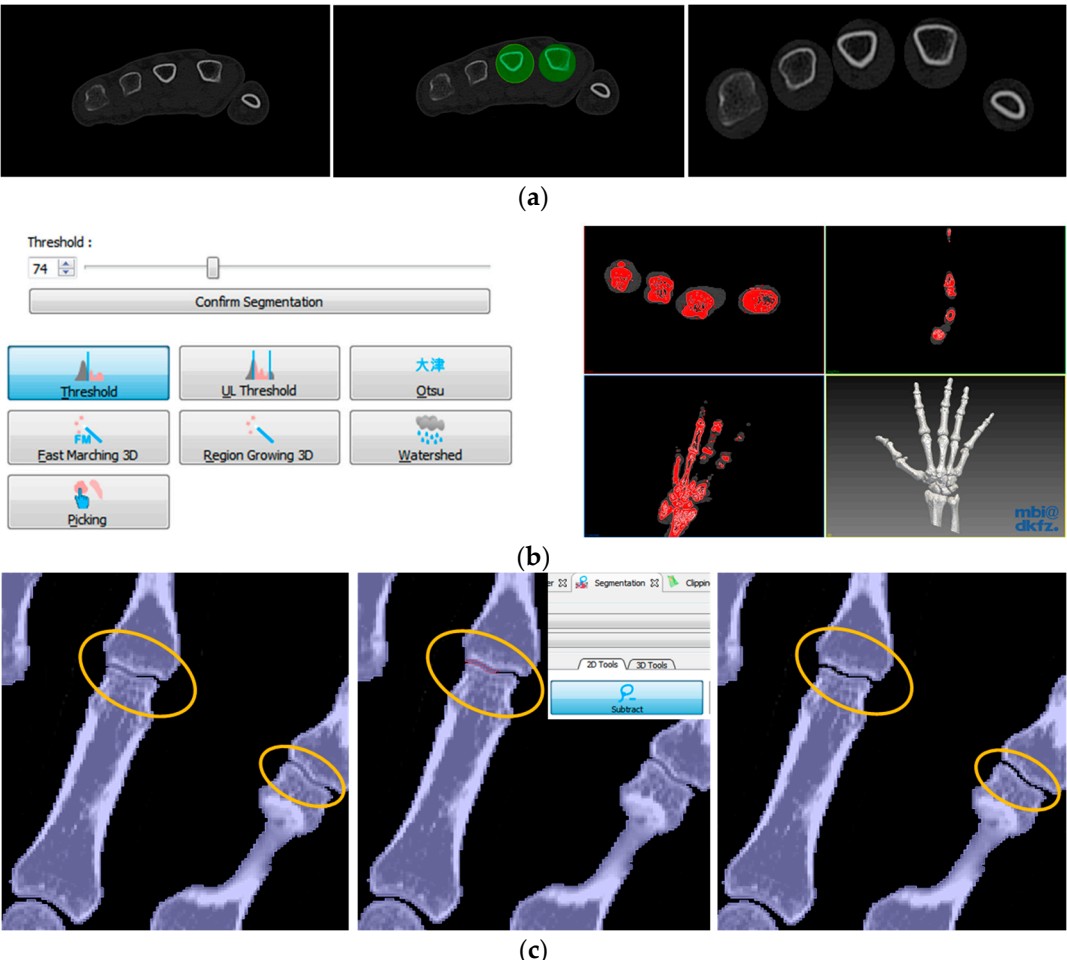

**Figure 3.** Reconstruction of 3D hand skeleton from a computed tomography (CT) scan: (**a**) Masking the hand bones to remove the hand skin, left: original CT scan in an axial view, middle: masking the bones using a scalable sphere, right: masked CT scan; (**b**) Segmentation of the hand skeleton from the masked CT scan using a threshold-based segmentation method in the Medical Imaging Interaction Toolkit (MITK); (**c**) Separation of the connected hand bones using a subtraction function in the Medical Imaging Interaction Toolkit (MITK), left: connected hand bones, middle: separation of the hand bones connected, and right: separated hand bones.

To ensure all the hand skeletons at different postures have the same bone surfaces, the 29 bones in the hand skeleton of posture 1 (template posture) were registered to those in the hand skeletons of the other nine postures (target postures), as shown in Figure 4. First, each bone in the template posture was roughly registered to that in a target posture by aligning three points selected at a similar position, each from the template bone surface and the target bone surface (Figure 5a,b). Then, a fine registration (Figure 5c) was performed to precisely register the template bone to the target bone using the ICP algorithm [18]. The registration of the template bone to the target bone was conducted in RapidForm 2006. Table 1 shows the mean registration accuracy of the bones from posture 1 to each of the other nine postures measured by root mean square error (RMSE) of Hausdorff distance [21]. The registered hand skeletons of the other nine postures and the hand skeleton of posture 1 having the same bone surfaces were then used to find finite finger joint CoRs.

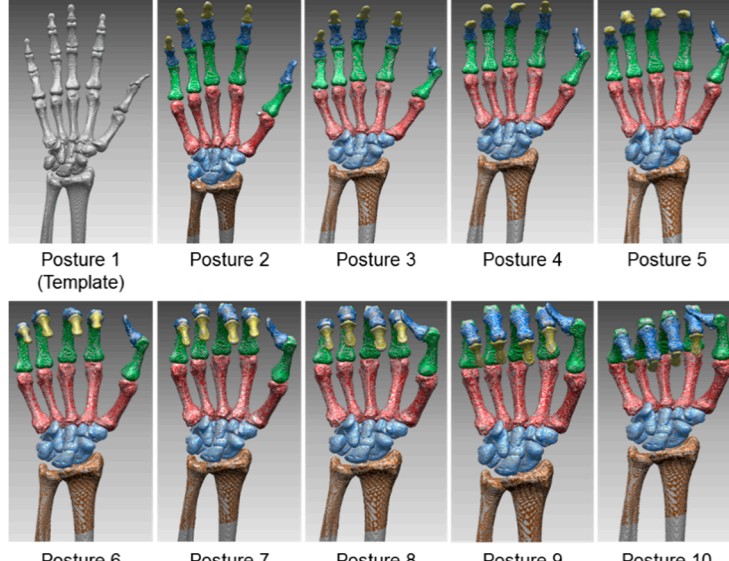

**Figure 4.** Registration of each bone in the hand skeleton of posture 1 (template posture) to the corresponding bones in the hand skeletons of the other nine postures.

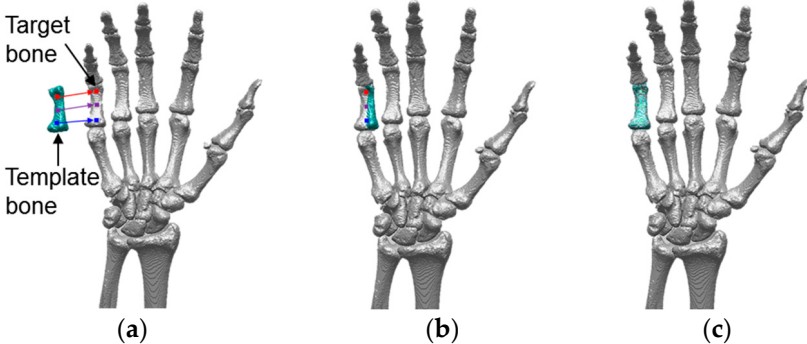

**Figure 5.** Registration of a bone in the template posture to that in a target posture: (**a**) Selection of three points at a similar position, each from the template bone surface and the target bone surface; (**b**) Initial registration by aligning the selected points; (**c**) Fine registration.

**Table 1.** Mean (± S.D.) registration accuracy of the hand skeletons (*n* = 15) in the template posture (posture 1) to those in the other postures (postures 2 to 10) measured by root mean square error (RMSE) of Hausdorff distance (unit: mm).

| Posture | RMSE | Posture | RMSE | Posture | RMSE |
|---------|------|---------|------|---------|------|
| 2 | 0.001 (± 0.001) | 5 | 0.002 (± 0.001) | 8 | 0.002 (± 0.001) |
| 3 | 0.002 (± 0.002) | 6 | 0.002 (± 0.001) | 9 | 0.001 (± 0) |
| 4 | 0.001 (± 0) | 7 | 0.002 (± 0.003) | 10 | 0.002 (± 0.001) |

## 2.3. Estimation of Finite Finger Joint CoRs

For the estimation of the finite CoRs of a finger joint, proximal bone surfaces of the joint at different postures were aligned to measure the relative motions of the distal bone of the joint from one posture to another, as illustrated in Figure 6. The alignment of the proximal bone surfaces of the joint was performed using the above-mentioned 3-point alignment method in RapidForm 2006. The proximal bone surfaces of the joint were exactly aligned with each other since the bone surfaces were the same.

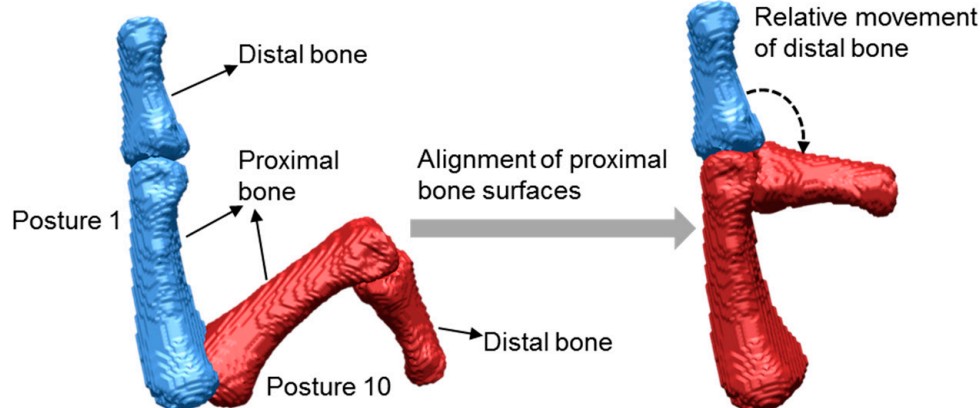

**Figure 6.** Alignment of the proximal bone surfaces of a joint at two postures to measure the relative motion of the distal bone of the joint between the two postures.

After the proximal bone surfaces were aligned, the centerlines of the distal bone surfaces of the joint at different postures were identified. First, a preliminary centerline of the distal bone surface at a posture was obtained using the principal component analysis (PCA) method. The PCA method consists of four steps: (1) calculation of the mean value of each of the three coordinates of all the vertices at the distal bone surface and subtraction of the corresponding mean value from each of the three coordinates of all the vertices to obtain new coordinates for the vertices, (2) calculation of a covariance matrix for the new coordinates, (3) calculation of the eigenvectors and eigenvalues of the covariance matrix, and (4) sorting the eigenvalues in a descending order to obtain the largest eigenvalue. The preliminary centerline of the distal bone surface was then identified as the line going through the axis determined by the corresponding eigenvector of the largest eigenvalue. Then, the fine centerline (Figure 7) of the distal bone surface was obtained by performing a linear fit of the centroids of the surface vertices at the perpendicular planes to the preliminary centerline along the distal bone shaft, the part of a bone excluding the two ends of the bone.

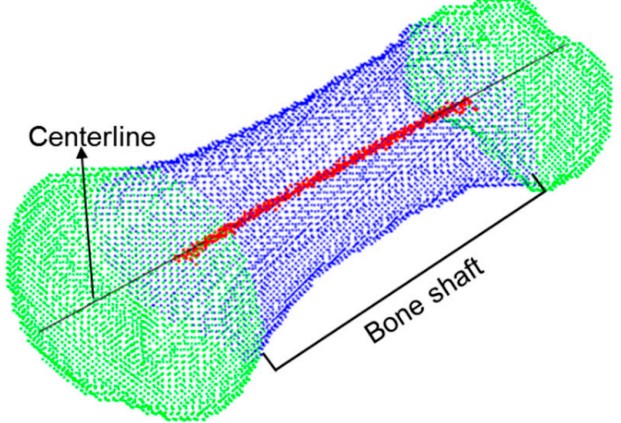

**Figure 7.** Identification of the centerline for the distal bone of a joint by performing a linear fit of the centroids (red) of the surface vertices at the perpendicular planes to a preliminary centerline along the distal bone shaft.

The finite finger joint CoR from one posture to another was determined as the intersection point of the centerlines of the distal bone surfaces from the two postures, as shown in Figure 8. A program was coded by Matlab R2017b (MathWorks, Inc., Natick, MA, USA) to perform the procedure of finding the centerlines of the distal bone surfaces and finite finger joint CoRs.

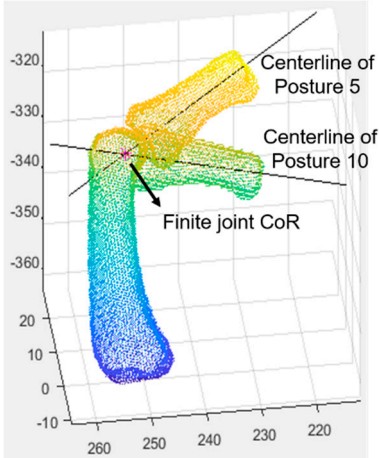

**Figure 8.** Determination of the finite finger joint center of rotation (CoR) between two postures by the intersection of the centerlines of the distal bone surfaces of the two postures.

*2.4. Evaluation of the Proposed Method*

The proposed method was compared with Reuleaux's method in terms of variation of estimated finite joint CoRs among different postures measured by mean distance among the estimated finite joint CoRs:

$$\text{Variance} = \frac{\sum_{\substack{i,\,j\,=\,1 \\ i\,\neq\,j}}^{N} d\big(\text{CoR}_i, \text{CoR}_j\big)}{{}_N C_2} \tag{1}$$

where $d\big(\text{CoR}_i, \text{CoR}_j\big)$ is the Euclidean distance between $\text{CoR}_i$ and $\text{CoR}_j$; $N$ is the number of estimated CoRs among different hand postures for a joint. Due to the relatively small size of a finger joint, its finite joint CoRs among different postures should not be far away from each other unless some mistakenly estimated finite joint CoRs appear. Therefore, the smaller the variation of estimated finite joint CoRs among different postures, the better the performance of the estimation method. As presented in Figure 9, Reuleaux's method estimates the finite joint CoR as the point of intersection of two lines that are the mid-perpendiculars of two distinct landmark displacement vectors [15].

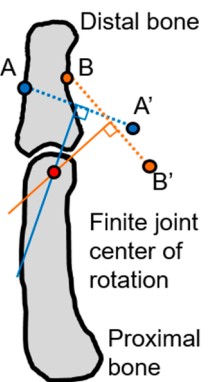

**Figure 9.** Determination of finite finger joint center of rotation (CoR) between two postures by Reuleaux's method.

**3. Results**

The locations of finite joint CoRs among different postures for the MCP, PIP, and DIP joints of the index, middle, ring, and little fingers of each participant were estimated with the proposed method and Reuleaux's method. For example, the estimated finite joint CoRs of the PIP joint among

different postures for the index finger are shown in Figure 10. No apparent errors in the locations of the estimated finite joint CoRs were found in the proposed method by visual inspection.

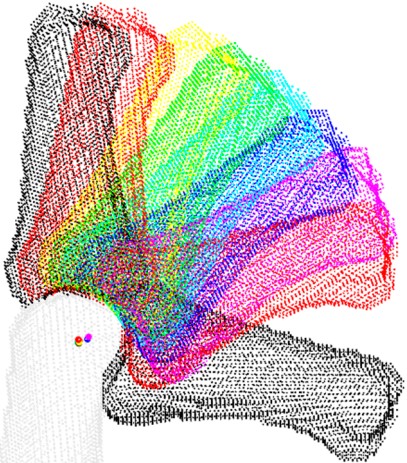

**Figure 10.** Estimated finite finger joint centers of rotation among different postures by the proposed method for the proximal interphalangeal joint of the index finger.

As shown in Figure 11, the proposed method achieved mean distances (0.5 mm to 1.4 mm) among the estimated finite joint CoRs at the MCP and DIP joints of all the four fingers and the PIP joints of the index, middle, and little fingers less than Reuleaux's method (0.6 mm to 3.9 mm); statistically significant differences were found at the MCP joint of the index finger ($t(14) = -2.24$, $p = 0.042$), the PIP joint of the index finger ($t(14) = -2.46$, $p = 0.028$), and the DIP joints of the index ($t(14) = -2.34$, $p = 0.035$), middle ($t(14) = -2.26$, $p = 0.040$), ring ($t(14) = -2.55$, $p = 0.023$), and little ($t(14) = -3.90$, $p = 0.002$) fingers. The proposed method showed a mean distance slightly higher than Reuleaux's method at the PIP joint of the ring finger but not statistically significant ($t(14) = 0.55$, $p = 0.594$).

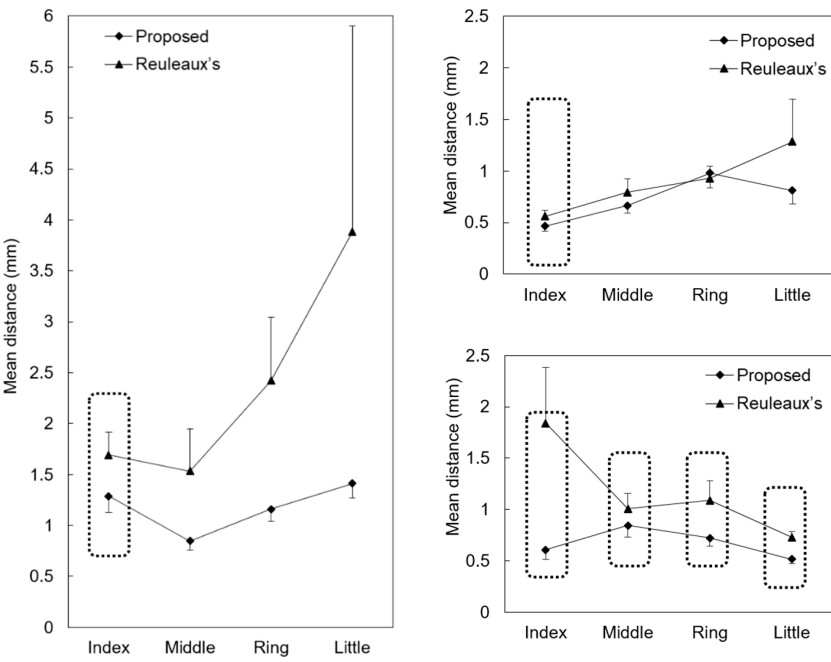

**Figure 11.** Mean distances and standard errors among the estimated finite joint centers of rotation by the proposed method and Reuleaux's method at the metacarpophalangeal (left), proximal interphalangeal (upper right), and distal interphalangeal (lower right) joints of the index, middle, ring, and little fingers (dotted box: significant difference at $\alpha = 0.05$).

To examine the effect of rotation angle on estimated finite joint CoRs, the mean distance between the finite joint CoR estimated under a rotation angle and other finite joint CoRs estimated under the remaining rotation angles was calculated. Rotation angles were divided into 18 groups every 5 degrees from 0 to 90 degrees. Then, the average value of the mean distances within each rotation group for all the participants was calculated and plotted. As shown in Figure 12, the average of mean distances of an estimated finite joint CoR to other estimated finite joint CoRs decreased rapidly as the rotation angle increased from 0–5-degree group to 5–10-degree group. Then, the decrease slowed down as the rotation angle increased from the 5–10-degree group to the 20–25-degree group and, lastly, became leveled off after the rotation angle reached 25 degrees. Large fluctuations were observed at the MCP and DIP joints in Reuleaux's method, whereas no fluctuation occurred in the proposed method.

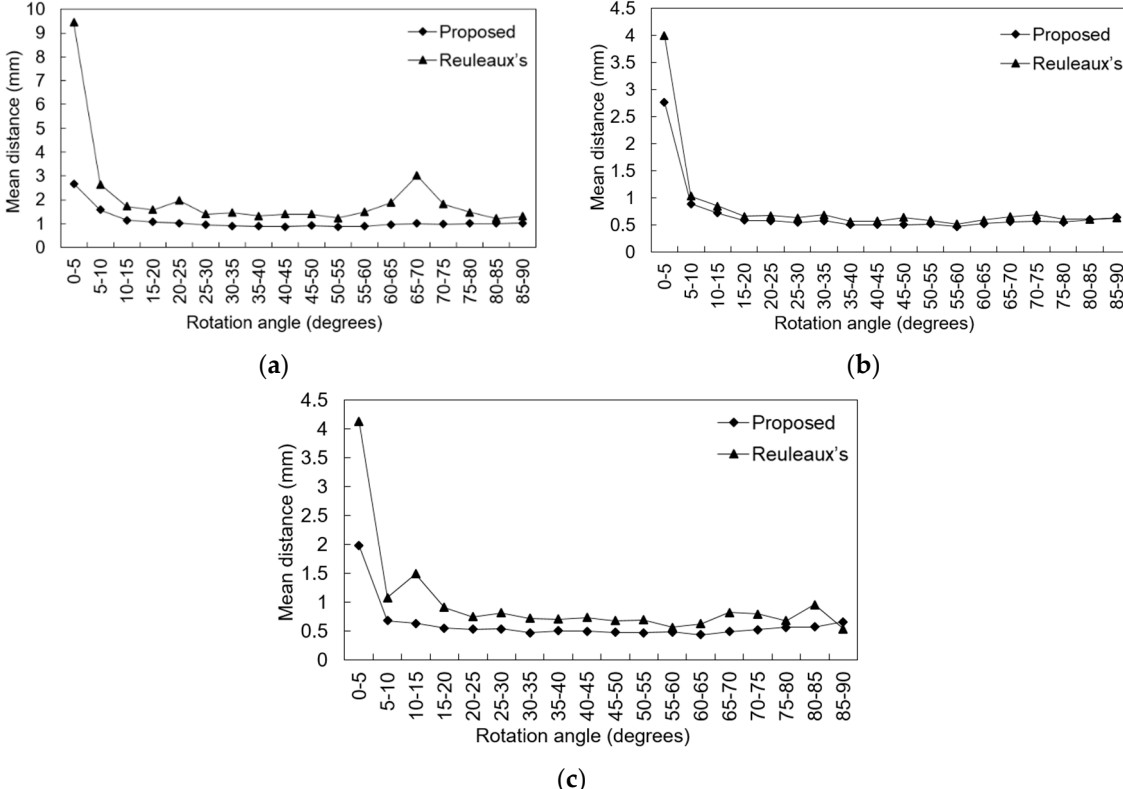

**Figure 12.** The average of mean distances of an estimated finite joint CoR to other estimated finite joint CoRs at each rotation angle group for (**a**) the metacarpophalangeal joint, (**b**) the proximal interphalangeal joint, and (**c**) the distal interphalangeal joint.

## 4. Discussion

The present study proposed to use the same bone surfaces among different hand postures for the estimation of finite joint CoRs by registration of each bone segment of the template posture to the corresponding bone segment of the other postures. Reconstructed hand skeletons from different CT scans at different hand postures could have different bone surfaces even with the same segmentation parameters due to the differences in intensity values among different CT scans. Different centerlines of the same distal bone segment at different postures could be identified if different bone surfaces were used, which could cause errors during the estimation of finite joint CoRs. Obvious errors observed in Figueroa et al.'s study could be caused by the usage of different bone surfaces in their study. In the present study, the identified centerlines of the same distal bone segment at different postures were exactly the same after using the same bone surfaces among those postures. Unlike Figueroa et al.'s study, no apparent errors were observed in the present study.

The proposed method was found superior to Reuleaux's method in terms of variation of estimated finite joint CoRs among different postures at most finger joints. The proposed method improved variation in estimated finite hand joint CoRs by 16.0% to 67.0% at different joints compared to Reuleaux's method. Reuleaux's method showed a much higher variation than the proposed method when the rotation angle was smaller than five degrees, as shown in Figure 12. Especially for the MCP joint, the variation in Reuleaux's method reached 9.4 mm, which is quite large as the sizes of the hand joints are small, whereas the variation in the proposed method achieved 2.7 mm. In Challis' study, it was concluded that Reuleaux's method could not perform well when the rotation angle was less than five degrees (mean error > 11 mm), which could explain why Reuleaux's method showed a large variation when the rotation angle was smaller than five degrees in the present study. The proposed method showed a slightly higher variation than Reuleaux's method (difference = 0.05 mm) at the PIP joint of the ring finger, which was caused by a larger variation at the PIP joint of the ring finger for the proposed method (2.6 mm) than Reuleaux's method (1.2 mm).

In the present study, the variation of estimated finite joint CoRs decreased as the rotation angle increased, which is in agreement with previous studies [15,22,23]. The accuracy of Reuleaux's method is dependent on the distribution of landmarks, whereas the proposed method does not need any landmarks. Furthermore, Reuleaux's method was found more sensitive to rotation angle than the proposed method. The differences in variation of estimated finite joint CoRs between the rotation angle group from 0 to 5 degrees to the remaining groups for Reuleaux's method and the proposed method for the MCP, PIP, and DIP joints were 6.1 mm, 1.1 mm, and 1.9 mm, respectively. Therefore, the proposed method outperformed Reuleaux's method, especially in the small rotation angle range (0–5 degrees). Besides, the fluctuations were found in variation among different rotation angle groups in Reuleaux's method but not in the proposed method, indicating that our method is more stable than Reuleaux's method.

The proposed method is similar to Figueroa et al.'s method. Both the methods estimate a finite finger joint CoR as the intersection point of the centerlines of the distal bone segments of a joint between two hand postures that are reconstructed from CT scans. The difference between the methods is that the proposed method includes an extra registration procedure to register each bone segment in the template posture to that in each of the nine remaining postures. By doing so, the ten postures can share the same bone surfaces so that the centerlines of the same distal bone segment identified at different postures are exactly the same to increase accuracy in estimation of finite CoRs. The computational complexity of the proposed method is similar to Figueroa et al.'s method. The extra registration procedure in the proposed method increases computation time, not computational complexity because Figueroa et al.'s method also includes the registration of the proximal bone segments. Next, Reuleaux's method estimates a finite joint CoR as the point of intersection of the mid-perpendiculars of two distinct landmark displacement vectors [15]. The landmarks are usually marked on the skin surface of a participant. The performance of Reuleaux's method depends on the number of landmarks, the distribution of the landmarks, skin deformation during rotation, and the range of rotation. The proposed method does not require any landmark and is less sensitive to the range of rotation. In this research, we put landmarks on bone surfaces by marking two distinct vertices to avoid the effect of skin deformation in Reuleaux's method. The proposed method needs to identify the centerlines of the distal bone segments, which could increase its computational complexity compared to Reuleaux's method.

Finite CoR indicates the CoR measured from a single finite displacement. The finite CoR of a human joint varies during rotation because the human joint is not an idealized spherical or axial joint [24]. That explains why the variation of the estimated finite CoRs of a finger joint between different hand postures occurred in our findings. Challis [15] reported that the mean errors in the estimation of the finite CoR were from 2 mm to 15 mm in his study. The moment arm of a human muscle to the finite CoR can only be 15 mm [25]; in this case, the error magnitude in Challis' study can be significant, and thus, the correct estimation of finite CoRs is important.

In the present study, the hand of a participant could be exposed to 5 mSv to 40 mSv (0.5 mSv to 4.0 mSv per scan trial × 10 scan trials) by CT scanning. Ionizing Radiation Regulations recommend a dose limit of 500 mSv for the hand in a year [26]. To protect the participant from radiation during scanning, they were covered by a lead-free radiation shielding apron and asked to lie face down and stretch out the right arm over the head to keep the rest of the body away from the CT scanner. Magnetic resonance imaging (MRI) that has no ionizing radiation is an alternative to CT scan. However, the MRI scan of the hand for one hand posture typically takes approximately 30 min; the participants need to stay still for 30 min; otherwise, the scan will fail. Furthermore, 10 postures used in this research need to be sequentially scanned in a short time period to guarantee the natural motion of the hand. It would take about 5 h to scan 10 postures using MRI, whereas it took less than one minute (3 to 5 s per scan) to scan 10 postures by CT. Furthermore, the CT scan performs better in bony structure imaging and is two times cheaper than MRI that is more suitable for soft tissue imaging. After carefully considering the various aspects mentioned above, a CT scan was selected in this research.

The present study did not include female participants due to the potential harm from radiation exposure to women during a CT scan. For future study, female participants can be included by using MRI without exposure to radiation. This research only included participants in their 20s. To study the effect of age on the estimation of finite finger joint CoRs, participants from different age groups need to be considered for future study. This research only estimated finite joint CoRs of the MCP, PIP, and DIP joints of the index, middle, ring, and little fingers. The applicability of the proposed method for the estimation of finite joint CoRs of other body joints needs to be explored. This research only studied a single hand motion from full extension to forming a fist. More types of hand motions need to be included to comprehensively study the effect of different hand motions on the estimation of finite joint CoRs. In a future study, hand link models will be established based on the results of the present study. The proposed method can be applied to examine joint movement for clinical applications, ergonomic product design, and robotic prosthetics, which require accurate and stable estimation of finite joint CoRs.

## 5. Conclusions

The present study proposed a novel method to estimate finite finger joint CoRs using 3D hand skeleton motions reconstructed from CT scans. After a hand skeleton was reconstructed from CT data for each of the ten postures, each bone segment in the template posture was registered to that in each of the nine remaining postures so that all the ten postures shared the same bone surfaces. Then, the proximal bone segment of a joint from each of the nine remaining postures was registered to that in the template posture. The centerline of the distal bone segment of the joint from each of the 10 postures was identified by the PCA method. The finite CoR of the joint between two different postures was lastly derived as the intersection point of the identified centerlines of the distal bone segments of the joint from the two postures.

The present study proposed to use the same bone surfaces among different hand postures for accurate estimation of finite finger joint CoRs. In Figueroa et al.'s method, different bone surfaces were used, which could cause errors in the estimated finite CoRs reported in their study. In contrast, no apparent errors in the estimated finite CoRs were observed in the proposed method after using the same bone surfaces.

The proposed method reduced the variation of estimated finite joint CoRs by 16.0% to 67.0% among the finger joints compared to Reuleaux's method. The variation of estimated finite joint CoRs decreased as the rotation angle of the joint increased for both the methods. Specifically, the proposed method significantly outperformed Reuleaux's method when the rotation angle was within 5 degrees.

The proposed method can be used to accurately assess human joint movement for various applications, such as biomechanical modeling, clinical applications, and ergonomic design. However, this research was limited by the use of a CT scan.

**Author Contributions:** Conceptualization, X.Y. and H.Y.; methodology: X.Y. and H.Y.; validation: X.Y. and H.Y.; formal analysis: X.Y., Z.L., and M.Z.; investigation: X.Y. and H.Y.; resources: X.Y., D.P., and H.Y.; data curation: X.Y., H.J., and H.Y.; writing—original draft preparation: X.Y. and Z.L.; writing—review and editing: X.Y., H.J., Y.H., D.P., and H.Y.; visualization: X.Y. and Z.L.; supervision: X.Y. and H.Y.; project administration: X.Y., H.J., and H.Y.; funding acquisition: X.Y. and H.Y. All authors have read and agreed to the published version of the manuscript.

**Funding:** This research was funded by the Fundamental Research Funds for the Central Universities (JUSRP12051) and the research programs of the National Research Foundation (NRF) of the Ministry of Education, Science, and Technology (2017M3C1B6070526; 2018R1A2A2A05023299; 2018K1A3A1A20026539) and the Ministry of Trade, Industry, and Energy (R0004840, 2020).

**Conflicts of Interest:** The authors declare no conflict of interest.

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
