# Peer review of "Estimation of Finite Finger Joint Centers of Rotation Using 3D Hand Skeleton Motions Reconstructed from CT Scans"

_applsci, doi:10.3390/app10249129_

Round 1

Reviewer 1 Report

The objective of this paper is the estimation of the hand centers of rotation. To this end, the authors develop an algorithm based on CT scans of fifteen voluntary participants. Ten hand postures are scanned (from a fully extended posture to a fist posture) and several processing steps are implemented to estimate the CORs of finger joints. The algorithm is compared to a different (Reuleaux’s) method and exhibits improved performance with respect to this latter.

The paper is interesting and quite well written. The topic is interesting and it has applications in rehabilitation, product design, and robotic prosthetics.

My main concern is related to the opportunity of using CT scans for this application,  as young subjects receive high dose radiation for this purpose. I understand that the study has been  approved by a Review Board.  However, I would like to know: -if this Board is equivalent to an Ethics Committee; -what dose of radiation has been employed for (10?) CT scans?; a discussion about the reasons which have led to the selection of CT and not the much safer RMN.

I cannot recommend publication unless the necessity of using radiations for this purpose is clearly stated.

As for pretty technical aspects, I believe that a short discussion about the main differences of the proposed algorithm with respect to state-of-the art may be useful, as long as a brief evaluation of the computational complexity, and possibly some details on possible applications (is this algorithm thought to be routinely employed for some application? Which one?)

Author Response

Thank you for the review on our manuscript. We appreciate your thoughtful comments and suggestions. We have edited the manuscript in response to the review comments. Specific responses to the comments are presented below and the revised portions of the manuscript are highlighted in yellow.

Reviewer 1

The objective of this paper is the estimation of the hand centers of rotation. To this end, the authors develop an algorithm based on CT scans of fifteen voluntary participants. Ten hand postures are scanned (from a fully extended posture to a fist posture) and several processing steps are implemented to estimate the CORs of finger joints. The algorithm is compared to a different (Reuleaux’s) method and exhibits improved performance with respect to this latter. The paper is interesting and quite well written. The topic is interesting and it has applications in rehabilitation, product design, and robotic prosthetics.

  1. My main concern is related to the opportunity of using CT scans for this application, as young subjects receive high dose radiation for this purpose. I understand that the study has been approved by a Review Board.  However, I would like to know: -if this Board is equivalent to an Ethics Committee; -what dose of radiation has been employed for (10?) CT scans?; a discussion about the reasons which have led to the selection of CT and not the much safer RMN. I cannot recommend publication unless the necessity of using radiations for this purpose is clearly stated.
  • The Institutional Review Board of Pohang Stroke and Spine Hospital which reviewed the protocol of the present study is equivalent to an Ethics Committee. The information related to the dose of radiation for hand CT scans and the reasons of using CT scans have been added in the discussion section.

  1. As for pretty technical aspects, I believe that a short discussion about the main differences of the proposed algorithm with respect to state-of-the art may be useful, as long as a brief evaluation of the computational complexity, and possibly some details on possible applications (is this algorithm thought to be routinely employed for some application? Which one?)
  • As the reviewer suggested, a paragraph which discusses the main differences of the proposed algorithm with the state-of-the art and a brief evaluation of computational complexity.

Reviewer 2 Report

This is an interesting paper presenting that CoR in the hand changes during joint rotation and therefore it is not constant. The findings can be used to explain the biomechanics of the hand, in robotics, kinematic analyses, simulation, etc. 

Major concerns:

The paper is well structured; however, I think, some of the text in the results section should be moved to the discussion section, since authors already discuss the results and compare them to other authors. Results should only provide plain results/data. Hence discussion will be and should be expanded.

Concerning the results, the authors should also explain why does the CoR move during joint rotation and why is it crucial that the CoR are estimated correctly at all joint rotations. Thu would be highly beneficial to the readers. I suggest the authors include a paragraph or special subsection. Additionally, the theory, why this happens can be used to discuss the obtained results by the authors.

I would also prefer to see a conclusion section. Although this is not mandatory by the guidelines of the Journal. The paper would hereby provide the main findings and show their importance. However, I let this decision to the editor/authors themselves.

Specific remarks:

Introduction

Figure 1 – figure clarity/resolution should be improved.

Material and methods

“Fifteen males (age = 23.7 ± 2.0 years, ranged from 20 to 28) with various hand sizes participated in the study.” – do you think age has an influence on the COR?

“The ten postures (postures 1 to 10; Figure 2) were selected from a natural hand motion starting from a fully extended posture (posture 1) and ending at a fist posture (posture 10) with a difference of approximately 10 degrees in flexion between two adjacent postures.” – how did you define/measure the 10 degrees while performing the CT?

“Table 1. This is a table. Tables should be placed in the main text near to the first time they are cited.” – remove the template text

“Though the hand skeletal structure is a non-rigid linkage system, the finite joint CoRs of a finger joint among different postures should not be far away from each other since the hand skeletal structure is usually assumed to be a rigid linkage system [11].” – vague sentence, please rephrase

Discussion

“Reconstructed hand skeletons from different CT scans at different hand postures could have different bone surfaces even with the same segmentation parameters due to the differences of intensity values among different CT scans.” – How does this influence your study? Does using the same bone surfaces eliminate this problem entirely? Discuss in the manuscript.

Author Response

Thank you for the review on our manuscript. We appreciate your thoughtful comments and suggestions. We have edited the manuscript in response to the review comments. Specific responses to the comments are presented below and the revised portions of the manuscript are highlighted in yellow.

Reviewer 2

This is an interesting paper presenting that CoR in the hand changes during joint rotation and therefore it is not constant. The findings can be used to explain the biomechanics of the hand, in robotics, kinematic analyses, simulation, etc.

Major concerns:

  1. The paper is well structured; however, I think, some of the text in the results section should be moved to the discussion section, since authors already discuss the results and compare them to other authors. Results should only provide plain results/data. Hence discussion will be and should be expanded.
  • As the reviewer suggested, we have moved some of the text in the Results section to the Discussion section.

  1. Concerning the results, the authors should also explain why does the CoR move during joint rotation and why is it crucial that the CoR are estimated correctly at all joint rotations. Thu would be highly beneficial to the readers. I suggest the authors include a paragraph or special subsection. Additionally, the theory, why this happens can be used to discuss the obtained results by the authors.
  • As the reviewer suggested, a paragraph has been added in the discussion section to explain why the CoR moves during joint rotation and the importance of correct estimation of CoR.

  1. I would also prefer to see a conclusion section. Although this is not mandatory by the guidelines of the Journal. The paper would hereby provide the main findings and show their importance. However, I let this decision to the editor/authors themselves.
  • As the reviewer suggested, a conclusion section has been added.

Specific remarks:

  1. (Introduction) Figure 1 – figure clarity/resolution should be improved.
  • As the reviewer suggested, the resolution of Figure 1 has been improved.

  1. (Material and Methods) “Fifteen males (age = 23.7 ± 2.0 years, ranged from 20 to 28) with various hand sizes participated in the study.” – do you think age has an influence on the COR?
  • We do not think age is a significant factor in CoR estimation; however, we have added the age issue as a further study to the discussion section.

  1. (Material and Methods) “The ten postures (postures 1 to 10; Figure 2) were selected from a natural hand motion starting from a fully extended posture (posture 1) and ending at a fist posture (posture 10) with a difference of approximately 10 degrees in flexion between two adjacent postures.” – how did you define/measure the 10 degrees while performing the CT?
  • The method of hand posture control while performing CT scanning has been described in the Materials and Methods section.

  1. (Material and Methods) “Table 1. This is a table. Tables should be placed in the main text near to the first time they are cited.” – remove the template text
  • The template text has been removed.

  1. (Material and Methods) “Though the hand skeletal structure is a non-rigid linkage system, the finite joint CoRs of a finger joint among different postures should not be far away from each other since the hand skeletal structure is usually assumed to be a rigid linkage system [11].” – vague sentence, please rephrase
  • The sentence has been rephrased.

  1. (Discussion) “Reconstructed hand skeletons from different CT scans at different hand postures could have different bone surfaces even with the same segmentation parameters due to the differences of intensity values among different CT scans.” – How does this influence your study? Does using the same bone surfaces eliminate this problem entirely? Discuss in the manuscript.
  • Figueroa et al.’s method uses different bone surfaces for CoR estimation, resulting in obvious errors in the estimated CoRs in their report. In contrast, our proposed method uses the same bone surfaces by registering the bone segments in the template posture to those in the remaining postures, resulting in no obvious errors of the estimated CoRs in our study. A paragraph which discusses the differences of Figueroa et al.’s method and our proposed method has been added in the manuscript.

Round 2

Reviewer 1 Report

The authors have properly taken into account my comments; hence I believe that the paper can be published (please spell-check the manuscript as a few typos are indeed present)

Author Response

Reviewer 1

  1. The authors have properly taken into account my comments; hence I believe that the paper can be published (please spell-check the manuscript as a few typos are indeed present)
  • The typos in the manuscript have been corrected and their corrections are highlighted in yellow.